# COVID-19 vaccine acceptability, and uptake among people living with HIV in Uganda

**Richard Muhindo**[1]*, **Stephen Okoboi**[2], **Agnes Kiragga**[2], **Rachel King**[3], **Walter Joseph Arinaitwe**[2], **Barbara Castelnuovo**[2]

1 Department of Nursing, College of Health Sciences, Makerere University, Kampala, Uganda, 2 Infectious Diseases Institute, College of Health Sciences, Makerere University, Kampala, Uganda, 3 University of California, San Francisco, California, United states of America

* r.muhindo@yahoo.com

## Abstract

### Background

Despite being a priority population for COVID-19 vaccination, limited data are available regarding acceptability of COVID-19 vaccines among people living with HIV (PLWH) in Sub-Saharan Africa. We described COVID-19 vaccine acceptability and factors associated with vaccine acceptability among PLWH in Uganda.

### Methods

This was a cross-sectional study conducted among PLWH, aged ≥18 years, enrolled participants who were seeking HIV care from six purposely selected accredited ART clinics in Kampala. We obtained data on vaccine acceptability defined as willingness to accept any of the available COVID-19 vaccines using interviewer-administered questionnaires. In addition, we assessed vaccination status, complacency regarding COVID-19 disease, vaccine confidence, and vaccine convenience. Factors associated with COVID-19 vaccine acceptability were evaluated using modified Poisson regression with robust standard errors.

### Results

We enrolled 767 participants of whom 485 (63%) were women. The median age was 33 years [interquartile range (IQR) 28–40] for women and 40 years [IQR], (34–47) for men. Of the respondents 534 (69.6%,95% confidence interval [CI]: 66.3%-72.8%) reported receiving at least one vaccine dose, with women significantly more likely than men to have been vaccinated (73% vs. 63%; $p = 0.003$). Among the unvaccinated 169 (72.7%, 95% CI: 66.6%-78.0%) were willing to accept vaccination, had greater vaccine confidence (85.9% had strong belief that the vaccines were effective; 81.9% that they were beneficial and 71% safe for PLWH; 90.5% had trust in health care professionals or 77.4% top government officials), and believed that it would be easy to obtain a vaccine if one decided to be vaccinated (83.6%). Vaccine acceptability was positively associated with greater vaccine confidence (adjusted prevalence ratio [aPR] 1.44; 95% CI: 1.08–1.90), and positive perception that it would be easy to obtain a vaccine (aPR 1.57; 95% CI: 1.26–1.96).

**Data Availability Statement:** All relevant data are within the manuscript and its Supporting Information files.

**Funding:** This study was supported through grant number D43TW009771 by National Institute for Health (NIH), Fogarty International Center.

**Competing interests:** The author(s) declare that they have no competing interests.

## Conclusion

vaccine acceptance was high among this cohort of PLWH, and was positively associated with greater vaccine confidence, and perceived easiness (convince) to obtained the vaccine. Building vaccine confidence and making vaccines easily accessible should be a priority for vaccination programs targeting PLWH.

## Background

Despite the current global decline in new infections, Coronavirus disease-2019 (COVID-19) still poses serious socio-economic, and health threats [1, 2]. As of 27th May2022, there had been more than 525 million reported infections with the severe acute respiratory coronavirus-2 (SARS CoV-2), the novel coronavirus that causes COVID-19, and more than 6.2 million reported deaths globally [3]. Over 8.9 million cases and 170,471 deaths had been reported in Africa, of which 164,366 cases and over 3,602 deaths in Uganda during the same period [3]. However, both cases, and deaths are likely to be underreported. Poor COVID-19 related health outcomes are substantial among high-risk persons, such as PLWH, diabetes mellitus, and cardiovascular diseases among others [4–8]. Compared to HIV-negative individuals, PLWH had a higher risk of SARS CoV-2 infection ([risk ratio (RR) 1.24, 95% confidence interval (CI), 1.05–1.46], and mortality (RR 1.78, 95% CI 1.21–2.60) [9]. COVID-19 vaccines have been shown to reduce infection severity and prevent deaths [10]. The increased risk of severe COVID-19 makes vaccination a priority for PLHIV, however concerns about adverse side effects, and negative impact on progression of HIV or antiretroviral therapy (ART) have been reported [11].

Vaccine acceptance defined by the degree to which individuals accept, question, or refuse vaccination [12], was already a global concern prior to the COVID-19 pandemic [13, 14]. Regional variations in COVID-19 vaccine acceptance have been reported in studies conducted mainly in Europe, North America, and Asia. In Canada, one survey found that compared to HIV-negative individuals, PLWH had lower intentions to vaccinate (65.2% versus 79.6%) [15]. Low intentions to vaccinate were also reported in China, as only 57.2% of PLWH were willing to receive a COVID-19 vaccine [16]. In India, the prevalence of vaccine hesitancy was found to be 38% among PLWH [17]. In the USA, one study found high acceptability (72%) among PLHIV [18], however it was lower among PLWH who reside in rural areas or inject drugs [18].

Sub-Saharan surveys in Ethiopia, and Nigeria have reported high prevalence (66.3%, and 53.8% respectively) of vaccine hesitancy among PLWH [19, 20].

Since the launch of the vaccination rollout in March 2021, Uganda aims to vaccinate 70% of the population [21–23] but as of. April 2022, only 21.5% of the Uganda population are estimated to have received two-vaccine doses [24, 25]. Proactively identifying vulnerable populations with co-morbidities to be prioritized for vaccination, and conducting surveys to understand barriers to uptake are currently among the priority actions for improving vaccination uptake [21]. Vaccine acceptability is determined by three factors: complacency, convenience and confidence [14, 26]. Confidence refers to trust in the effectiveness, the safety of vaccines and the system that delivers them; complacency refers to low perceived risk of vaccine-preventable diseases where vaccination is not deemed a necessary preventive action; convenience is measured by the extent to which physical availability, affordability and willingness-to-pay, geographical accessibility, ability to understand (language and health literacy) affect acceptability [26]. However, Context-specific factors for vaccine acceptance are needed to inform strategies to promote vaccine

uptake [27]. Therefore, this study sought to describe COVID-19 vaccine acceptability, and uptake in adults living with HIV, in an urban setting in Uganda.

## Methods

### Study design and setting

Between January and April 2022, we conducted a cross-sectional survey among PLHIV at 6 public health facilities in the Kampala metropolitan area with 40,228 PLWH ≥ 15 years enrolled in care (Komamboga, Kisenyi, Kiswa, Kitebi, Kawaala, and Kasangati). For this study we approached PLWH ≥ 18 years seeking ART services regardless of the vaccination status who were able to speak English or Luganda (the local language in the area of Kampala).

### Recruitment and data collection procedures

To estimate the study sample size, we aimed to achieve a precision of 5%. We assumed vaccine acceptance of 50%, as no prior studies in the region had described PLWH with regard to COVID-19 vaccination. A total of 768 respondents were estimated (two-sided test at 95% level of significance, 5% margin of error, and a design effect of 2) using Cochran formula [28]. Altogether 40,228 PLHIV ≥ 15 years were enrolled in care at the study sites. Of these 4543 (11.3%) were at Komamboga, 11950 (29.7%) at Kisenyi, (14.1%) at Kiswa, 6802 (16.9%) at Kitebi, 8825 (21.9%) at Kawaala, and 2420 (6%) at Kasangati. Using proportion to size allocation, we enrolled respondents at each health centre, while waiting to be seen by the health care providers. Over 70 PLWH received care at each study site on a daily basis. Using simple random sampling, random numbers were given to potential respondents in the clinic waiting area on each recruitment day.

We obtained data on COVID-19 vaccine acceptability, and uptake using a questionnaire. The questionnaire was developed based on literature [26], reviewed by a multidisciplinary team of research experts (medical, statistics, and social sciences), and piloted among 15 PLWH at a non-participating facility. The questionnaire consisted of 26 question items (Cronbach's α coefficient, 0.79), that were used to assess the major independent variables. Five items to assess complacency (α = 0.67), Nine to assess perceived vaccine confidence (α = 0.74), one to assess willingness to vaccinate, and four to assess convenience (α = 0.43). The English questionnaire was administered through face-to-face interviews by experienced research assistants (RAs), in the language of the participant's preference at convenient venue within the health facility to ensure privacy and confidentiality. Prior to data collection, the research assistants received training on the protocol, and participated in the piloting of the questionnaire. All RAs were native speakers with experience of administering English interviews in the local language. Ethical clearance was obtained from the Infectious Diseases Institute Research Ethics Committee (IDIREC REF 036/2021), and the Uganda National Council for Science and Technology (HS HS1902ES). All respondents provided written informed consent in their language of preference.

### Study variables

The primary outcome variable was vaccine acceptability defined as willingness to accept any of the current available vaccines. Data was obtained on vaccination status defined as receipt of one or more vaccine doses (0 = No, and 1 = Yes), and unvaccinated respondents were asked if they will receive the vaccine (1 = 'Strongly disagree' to 5 = 'Strongly agree). The main independent variables were categorized as confidence, complacency, and convenience [26].*Confidence* was assessed by asking whether respondents perceived available vaccines to be safe and

effective, whether they trusted health care professionals, and political leaders regarding promoting uptake of the vaccines. Respondents were considered to have confidence if they strongly agreed or agreed that COVID-19 vaccines were safe, and effective for PLWH; trusted health professionals, government officers, and politicians promoting COVID-19 vaccination for PLWH. *Complacency* was assessed by asking respondents whether they thought they were at risk of contracting COVID-19, and whether they perceived COVID-19 vaccination beneficial in their circumstances. Respondents were considered complacent if they rated their risk of contracting COVID-19 to be low, and vaccination non-beneficial. *Convenience* on the other hand was assessed by asking respondents how easy or difficult it would be to get vaccinated if they desired to do so. Vaccination was considered convenient if respondents perceived obtaining vaccines to be easy. All item responses were measured on 5-point Likert scale (1 = 'strongly disagree' to 5 = 'strongly agree'). Likert scale (ordinal data) were dichotomized for confidence, convenience, complacency, and willingness to vaccinate. PLHIV were coded as "0, and 1" respectively as unwilling to accept the vaccine (strongly disagree/disagree/somewhat disagree), and willing to vaccinate (strongly agree/agree).

## Statistical analysis

We used proportions to describe respondents' demographic characteristics, their perceptions regarding COVID-19, information sources, vaccines, and willingness to vaccinate. Pearson's Chi square ($\chi^2$) test was used to examine if vaccination status, and acceptability varied by health facility, and gender. Since vaccine acceptability was high (prevalence >10) among the respondents, we evaluated associated factors using a modified Poisson regression model with robust standard errors [29]. Both deviance, and Pearson chi-square goodness of fit were conducted to asses model, and none was statistically significant. Crude and adjusted prevalence ratios (PR) and 95% confidence intervals (CI) were estimated. We considered two-sided p-values of 0.05 or less statistically significant. Analyses were completed using Stata version 15.0 (StataCorp, College Station, TX).

## Results

### Population characteristics

Analysis was performed on 767 PLWH (99.87%), of whom 485 (63.2%) were female. The median age was 36 years [interquartile range (IQR) 29–44], almost half (47.2%) were married 48.4% had obtained secondary or tertiary education, and 15.2% reported formal employment. One-third (33.2%) had ever taken a COVID-19 test, with 37/255 (14.5%) reporting positive results. Relatedly, 342 (44.6%) had a family member tested for COVID-19, with 136 (40%) reporting positive results (Table 1).

### COVID-19 vaccine uptake, and acceptability

Overall, 534 (69.6%,95% confidence interval [CI]: 66.3%-72.8% reported receiving at least one vaccine dose. Compared to women, men had lower vaccination uptake (73% vs.63%; p = 0.003), however, among the unvaccinated acceptance to vaccinate did not significantly vary between men and women (70.5% vs.74.5%; p = 0.49). Overall, among the unvaccinated, 169 (72.7%, 95% CI: 66.6%-78.0%) were willing to accept COVID-19 vaccination, while 92% among the vaccinated were willing to accept a booster dose. Vaccination uptake significantly varied across facilities (95.5% Kasangati, 79.8% Kiswa, 76.9 Kitebi,70.2% Kisenyi, 57.2% Kawaala, 54.5% Komambaga; p = 0.001). Results shown in Table 2.

**Table 1. Respondent characteristics (N = 767).**

| Variable | N (%) or median (IQR) |
|---|---|
| **Duration on ART (Months)** | 84 (48–144) |
| **Age (years)** | |
| 18–24 | 68 (8.9) |
| 25–35 | 296 (38.6) |
| 36–40 | 148 (19.3) |
| ≥ 41 | 255 (33.3) |
| **Sex** | |
| Male | 282 (36.8) |
| Female | 485 (63.2) |
| **Education level** | |
| None | 54 (6.0) |
| Primary | 342 (44.6) |
| Secondary | 295 (38.5) |
| Higher education | 76 (9.9) |
| **Marital status** | |
| Married | 362 (47.2) |
| Separated | 151 (19.7) |
| Widow | 73 (9.5) |
| Never married | 181 (23.6) |
| **Employment status** | |
| Formal employment | 116 (15.2) |
| Self-employed | 439 (57.2) |
| Informal employment | 105 (13.7) |
| Unemployed | 107 (13.9) |
| **Previously suffered from COVID-19** | |
| Not sure | 84 (10.9) |
| Surely not | 525 (68.5) |
| Probably | 104 (13.6) |
| Yes | 54 (7.04) |
| **Ever tested for COVID-19** | |
| No | 512 (66.8) |
| Yes-Negative | 218 (28.4) |
| Yes-Positive | 37 (4.82) |
| Family member ever tested | |
| No | 379 (49.4) |
| Yes-Negative | 206 (26.9) |
| Yes-Positive | 136 (17.7) |
| I don't know | 46 (6.0) |

ART: antiretroviral treatment

## Complacency regarding COVID-19 infection, and vaccine confidence

COVID-19 was perceived as a serious disease by 581 (75.8%), and 643 (85.1%) thought COVID-19 poses a health risk to people in Uganda. However, 450 (58.7%) reported not being worried about getting COVID-19. Nearly two-thirds (63%) rated their future risk of contracting COVID-19 to be low or none. Risk perception did not differ between vaccinated and unvaccinated (62.5% vs. 63.5%; p = 0.78). However, compared to unvaccinated PLWH,

**Table 2. COVID-19 vaccination acceptability.**

| Variable | N (%) | p-valve |
|---|---|---|
| **Received a COVID-19 jab per study site (yes)** | | **0.001*** |
| Kiswa HC (n = 109) | 87 (79.8) | |
| Komamboga HC (n = 88) | 48 (54.5) | |
| Kitebi HC (n = 130) | 100 (76.9) | |
| Kasangati HC (n = 46) | 44 (95.5) | |
| Kisenyi HC (n = 228) | 160 (70.2) | |
| Kawaala HC (n = 166) | 95 (57.2) | |
| Overall (N = 767) | 534 (69.6) | |
| **Received a COVID-19 jab, men Vs women (yes)** | | **0.003*** |
| Male (n = 282) | 178 (63) | |
| Female (n = 485) | 356 (73.4) | |
| **Willingness to receive a booster dose among those already vaccinated (N = 445)** | | NA |
| Yes | 409 (91.9) | |
| **willing to accept a COVID-19 jab (N = 233)** | | |
| Strongly disagree + disagree | 64 (27.3) | |
| Strongly agree + agree | 169 (72.7) | |
| **willing to accept a COVID-19 jab, men (N = 100) Vs women (N = 133)** | | 0.496 |
| Men | 71 (70.5) | |
| Women | 99 (74.4) | |

HC: Health Centre

vaccinated PLWH were more likely to believe they have some immunity against COVID-19. 72.5%; vs62.2% p = 0.005). Majority (85.9%) believed generally vaccines are beneficial, and effective in controlling diseases. COVID-19 vaccination for PLWH was perceived highly beneficial by 81.9% (628) for all PLWH, 640 (83.4%) agreed that vaccination reduces severe illness and death. However, less than half (330 or 43%) agreed that most Ugandans want to be vaccinated against COVID-19. Results shown in Table 3.

## Confidence in information sources, and convenience of vaccination program

Compared to government top officials, more agreed to trust all information regarding vaccination from a health care professional than those trusting government top officials (90.5%) versus 77.4%. Majority (80.3%) reported receiving information regarding COVID-19 vaccination during the clinic visit (Table 4). The most common trusted sources of information were: health professionals (24.1%), ministry of health (22.5%), president (17.9%), religious leaders (15.7%), and top government official (11.3%). Relatedly, 621 (81.0%) reported availability of COVID-19 vaccines, and 641 (83.6%) agreed it would be easy to get vaccinated if one decided. Overall, 80.8% were satisfied with the current handling of the vaccination exercise, and 85.1% would support government if vaccination would become mandatory.

## Associations with COVID-19 vaccination acceptability

In the multivariable model, the factors associated with vaccine acceptance were: attainment of secondary education (adjusted prevalence ratio [aPR] 0.72; 95% CI: 0.56–0.94), being in the age category of 36–40 years and 41–50 years compared to being in the age category 18–24 years (aPR 0.75; 95% CI: 0.57–0.98, and aPR 0.74; 95% CI: 0.56–0.98 respectively), positive

**Table 3. Complacency regarding COVID-19 infection, and vaccine confidence.**

| Variable | N (%) |
|---|---|
| **How serious is contracting COVID-19 as a disease** | |
| Not very serious + not serious | 186 (24.3) |
| Serious + very serious | 581 (75.8) |
| **Perceived risk of contracting COVID-19 in future** | |
| Not at all | 69 (9.0) |
| Slightly | 413 (53.8) |
| Very likely + extremely likely | 285 (37.2) |
| **Worry about getting COVID-19** | |
| Not very worried/not worried | 450 (58.7) |
| Very worried + extremely worried | 317 (41.3) |
| **If you contracted COVID-19, how big would it be as a health threat** | |
| No threat at all | 42 (5.5) |
| Minor | 457 (59.6) |
| Major | 268 (34.9) |
| **COVID-19 poses a risk to people in Uganda** | |
| Strongly disagree + disagree | 114 (14.9) |
| Strongly agree + agree | 653 (85.1) |
| **Have some immunity against COVID-19** | |
| Strongly disagree + disagree | 235 (30.6) |
| Strongly agree + agree | 532 (69.4) |
| **Believe vaccines are effective in controlling diseases** | |
| Strongly disagree +disagree | 104 (14.1) |
| Strongly agree + agree | 663 (85.9) |
| **Most people including children experience little or no side effects from vaccines** | |
| Strongly disagree +disagree | 259 (33.8) |
| Strongly agree + agree | 508 (66.2) |
| **Most Ugandans want to be vaccinated against COVID-19** | |
| Strongly disagree +disagree | 437 (57.0) |
| Strongly agree + agree | 330 (43.0) |
| **Important that all PLHIV should be vaccinated against COVID-19** | |
| Strongly disagree +disagree | 139 (18.1) |
| Strong agree + agree | 628 (81.9) |
| **All COVID-19 vaccines in Uganda reduce severe illness, and death** | |
| Strong disagree +disagree | 127 (16.6) |
| Strong agree + agree | 640 (83.4) |
| **Members of your family believe COVID-19 vaccines in Uganda are safe** | |
| Strongly disagree + disagree | 179 (23.3) |
| Strongly agree + agree | 588 (76.7) |
| **Your friends believe all COVID-19 in Uganda are safe** | |
| Strongly disagree + disagree | 374 (48.8) |
| Strongly agree + agree | 393 (51.2) |
| **Significant people like religious leaders, and politicians believe vaccines in Uganda are safe** | |
| Strongly disagree + disagree | 150 (19.6) |
| Strongly agree + disagree | 617 (80.4) |
| **All COVID-19 vaccines in Uganda are safe for PLWH** | |
| Strong disagree +disagree | 222 (28.9) |
| Strong agree + agree | 545 (71.1) |

(*Continued*)

**Table 3.** (Continued)

| Variable | N (%) |
|---|---|
| **Overall, how do you rate the benefit of COVID-19 vaccination among PLWH** | |
| Very harmful/harmful | 43 (5.6) |
| Neither harmful nor beneficial | 142 (18.5) |
| Beneficial | 582 (75.9) |

PLWH: People living with HIV

confidence that vaccination in general is beneficial (aPR 1.44; 95% CI: 1.08–1.90), COVID-19 vaccines are safe for PLWH (aPR 1.26; 95% CI: 1.06–1.51), and conveniently easy to obtain (aPR 1.57; 95% CI: 1.26–1.96) (Table 5).

**Table 4. Confidence in information sources, and convenience of vaccination program (N = 767).**

| Variable | N (%) |
|---|---|
| **Trust all information regarding COVID-19 vaccination provided by health care professionals** | |
| Strongly disagree + disagree | 73 (9.5) |
| Strongly agree + agree | 694 (90.5) |
| **Trust all information provided by top government officials** | |
| Strongly disagree + disagree | 173 (22.6) |
| Strongly agree + agree | 594 (77.4) |
| **Most trusted source of information** | |
| Celebrities | 25 (1.2) |
| Religious leaders | 339 (15.7) |
| President | 388 (17.9) |
| Top government official | 244 (11.3) |
| Health professionals | 521 (24.1) |
| Social media | 55 (2.5) |
| Friends | 88 (4.1) |
| MOH | 485 (22.5) |
| Cultural leader | 7 (0.3) |
| None | 8 (0.4) |
| **Government should force people to be vaccinated** | |
| Strongly disagree + disagree | 114 (14.9) |
| Strongly agree + agree | 653 (85.1) |
| **If you decided to get vaccinated, it would be easy to obtain the vaccine** | |
| Strongly disagree + disagree | 126 (16.4) |
| Strongly agree + agree | 641 (83.6) |
| **Currently vaccines are available at your nearest health facility** | |
| No | 146 (19.0) |
| Yes | 621 (81.0) |
| **During your clinic visit, you were provided information regarding vaccination** | |
| No | 151 (19.7) |
| Yes | 616 (80.3) |
| **You are satisfied with the current handling of the vaccination exercise** | |
| Strong disagree +disagree | 147 (19.2) |
| Strongly agree + agree | 620 (80.8) |

MOH: Ministry of health

**Table 5. Modified Poisson regression multivariable model for association of COVID-19 vaccine acceptance.**

| | Crude prevalence ratio (PR) | | | Adjusted prevalence ratio (PR) | | |
|---|---|---|---|---|---|---|
| | PR (SE) | 95% CI | p-value | PR (SE) | 95% CI | p-value |
| **Age category in years** | | | | | | |
| 18–24 | | | Reference | | | |
| 25–35 | 0.83 (.09) | 0.66–1.03 | 0.09 | 0.82 (.09) | 0.67–1.03 | 0.088 |
| 36–40 | 0.81 (.1) | 0.63–1.05 | 0.12 | 0.75 (.1) | 0.57–0.98 | **0.036**[*] |
| 41–50 | 0.77 (.01) | 0.59–1.01 | 0.06 | 0.74 (.1) | 0.56–0.98 | **0.036**[*] |
| 51 + | 0.98 (.05) | 0.76–1.25 | 0.87 | 0.78 (.01) | 0.61–1.02 | 0.072 |
| **Level of education** | | | | | | |
| None | | | Reference | | | |
| primary | 0.90 (.1) | 0.69–1.17 | 0.43 | 0.79 (.1) | 0.61–1.03 | 0.07 |
| secondary | 0.84 (.1) | 0.64–1.11 | 0.22 | 0.72 (.09) | 0.56–0.94 | **0.02**[*] |
| Higher education | 1.04 (.1) | 0.74–1.45 | 0.81 | 1.00 (.1) | 0.72–1.41 | 0.95 |
| **Marital status** | | | | | | |
| Married | | | Reference | | | |
| Separated | 0.98 (.1) | 0.79–1.21 | 0.88 | 1.02 (.09) | 0.86–1.22 | 0.79 |
| Widow | 1.03 (.1) | 0.77–1.38 | 0.43 | 0.99 (.05) | 0.75–1.31 | 0.98 |
| Never married | 1.08 (.1) | 0.89–1.30 | 0.40 | 0.97 (.05) | 0.78–1.14 | 0.58 |
| **Gender** | | | | | | |
| Male | | | | | | |
| Female | 1.05 (.08) | 0.90–1.24 | 0.50 | 1.01 (.07) | 0.87–1.17 | 0.83 |
| **Seriousness of COVID-19** | | | | | | |
| Not serious | | | Reference | | | |
| serious | 1.17 (.1) | 0.95–1.43 | 0.11 | 1.11 (.1) | 0.92–1.35 | 0.68 |
| **Perceived Risk** | | | | | | |
| Slight | | | Reference | | | |
| Very likely | 1.04 (.08) | 0.89–1.22 | 0.59 | 0.96 (.08) | 0.81–1.14 | 0.68 |
| **Personal health threat** | | | | | | |
| Minor | | | Reference | | | |
| Major | 1.23 (.09) | 1.06–1.42 | **0.005**[*] | 1.01 (.07) | 0.87–1.16 | 0.89 |
| **Prior COVID-19 testing** | | | | | | |
| No | | | Reference | | | |
| Yes | 1.06 (.09) | 0.89–1.27 | 0.47 | 1.04 (.03) | 0.94–1.29 | 0.23 |
| **General belief in benefit of vaccines** | | | | | | |
| Disagree | | | Reference | | | |
| Agree | 1.82 (.2) | 1.36–2.43 | **0.001**[*] | 1.44 (.2) | 1.08–1.90 | **0.01**[*] |
| **Most Ugandans want to be vaccinated** | | | | | | |
| Disagree | | | Reference | | | |
| Agree | 1.13 (.08) | 0.97–1.32 | 0.09 | 0.96 (.06) | 0.84–1.11 | 0.58 |
| **COVID-19 vaccination among PLWH** | | | | | | |
| Harmful | | | **Reference** | | | |
| Beneficial | 1.43 (.1) | 1.21–1.69 | **0.001**[*] | 1.08 (.09) | 0.91–1.29 | 0.37 |
| **COVID-19 vaccines in Uganda are safe for PLWH** | | | | | | |
| Not safe | | | Reference | | | |
| Safe | 1.59 (.1) | 1.35–1.87 | **0.001**[*] | 1.26 (.1) | 1.06–1.51 | **0.008**[*] |
| **Access to COVID-19 jabs** | | | | | | |
| Difficult | | | | | | |
| Easy | 1.83 (.2) | 1.44–2.33 | **0.001**[*] | 1.57 (.1) | 1.26–1.96 | **0.001**[*] |

Deviance goodness-of-fit = 21.51392, Prob > chi2(217) = 1.0000, Pearson goodness-of-fit = 20.46452, Prob > chi2(217) = 1.0000

Relatedly, the factors associated with vaccination uptake were: perception that vaccination was beneficial for PLWH (adjusted prevalence ratio [aPR] 1.18; 95% confidence interval [CI]: 1.04–1.33), vaccines were safe for PLWH (aPR 1.45; 95% CI: 1.24–1.70), easy to obtain the vaccine (aPR 1.64; 95% CI: 1.31–2.05), being unemployed (aPR 0.83; 95% CI: 0.69–0.99; p = 0.04), being female (aPR 1.17; 95% CI: 1.05–1.29), and prior testing for COVID-19 (aPR 1.19; 95% CI: 1.09–1.29) (Table 6).

## Discussion

This study sought to describe COVID-19 vaccine acceptability, and associated factors among PLWH in Uganda. Over two-thirds (72.7%) of the unvaccinated PLWH were willing to accept COVID-19 vaccines. Positive belief that vaccination is beneficial in general, confidence that vaccines were safe for PLWH, and belief that it would be conveniently easily to obtain the vaccine were positively associated with willingness to vaccinate. Vaccine acceptance was negatively associated with attainment of secondary education, and being in the age category of 36–50 years.

Our findings show high acceptability compared to those previously reported in SSA. Among PLWH, low willingness to accept vaccination were reported in Nigeria and Ethiopia (46.2%, and 33.7% respectively) [19, 20]. However, our findings are consistent with those reported in studies conducted in middle, and high income countries, where moderate to high acceptability [15, 16, 18, 30] (57%, 62%, 65, 70%, 72, and 80% respectively) were reported in China, India, Canada, Ireland, France, and Australia among PLWH [15–17, 30–32]. The vigorous government of Uganda campaign to promote COVID-19 vaccination could explain the high acceptability [21, 22].

We also observed that over two-thirds (69%) had received at least one dose of COVID-19 vaccine, suggesting high accessibility, and availability [14, 33]. However, men compared to women were less likely to have vaccinated (63% vs.73%) or willing to vaccinate (30% vs.25%). Men are known to less likely seek medical services than women in SSA) [34, 35]. According to the World Health organization, vaccine acceptance is influenced by a constellation of factors including confidence, complacency, and convenience [14, 26]. In this study acceptance was mainly explained by greater vaccine confidence [strong belief that the vaccines were effective (85.9%); beneficial (81.9%); safe (71%) for PLWH); trust of information sources (health care professionals, 90.5% or topical government officials,77.4%)], and belief that it would be easy to obtain a vaccine (convenience) if one decided to vaccinate (83.6%). Prior studies in China, Canada, India, and France among PLWH also reported lack of confidence in vaccine safety as barrier to vaccine acceptability [11, 15, 17, 31, 36]. These findings show the need for tailored messages to build vaccine confidence among PLHIV with emphasis. on COVID-19 vaccines and HIV, and COVID-19 vaccines and ART.

Prior literature suggests that PLWH with primary or no education were less likely to accept vaccination [11, 37]. In this study, PLWH with secondary education were less willing to accept vaccination (67% vs. 81.2%), but had slightly higher vaccination uptake (73.2% vs. 70.4%). Similarly, PLWH aged between 36–50 years were less willing to accept vaccination, but reported similar vaccination uptake with PLWH aged between 18–24 years (69% vs. 68.4%) This could be explained by the fact that vaccine uptake can still be high even where reluctance exists where vaccination is mandatory for one to access some services or travel [14, 38] or passive acceptance i.e. compliance by a public that accedes to recommendations, and social pressure [38].

The strength of our study includes: ours in the first study in Uganda to the best of our knowledge to provide insights about acceptability of COVID-19 vaccines among PLWH,

**Table 6. Modified Poisson regression multivariable model for association of COVID-19 vaccination uptake.**

| | Crude prevalence ratio (PR) | | | Adjusted prevalence ratio (PR) | | |
|---|---|---|---|---|---|---|
| | PR (SE) | 95% CI | p-value | PR (SE) | 95% CI | p-value |
| **Age category in years** | | | | | | |
| 18–24 | | | Reference | | | |
| 25–35 | 0.99 (.08) | 0.83–1.18 | 0.93 | 1.05 (.08) | 0.89–1.24 | 0.55 |
| 36–40 | 0.98 (.09) | 0.80–1.19 | 0.82 | 1.07 (.10) | 0.88–1.29 | 0.49 |
| 41–50 | 1.03 (.09) | 0.86–1.25 | 0.73 | 1.17 (.11) | 0.97–1.42 | 0.09 |
| 51 + | 1.06 (.10) | 0.87–1.29 | 0.56 | 1.17 (.13) | 0.94–1.46 | 0.15 |
| **Level of education** | | | | | | |
| None | | | Reference | | | |
| primary | 0.90 (.08) | 0.74–1.09 | 0.29 | 0.91 (.07) | 0.77–1.07 | 0.25 |
| secondary | 1.04 (.09) | 0.86–1.25 | 0.68 | 1.02 (.09) | 0.86–1.21 | 0.83 |
| Higher education | 1.18 (.12) | 0.96–1.44 | 0.11 | 1.06 (.10) | 0.88–1.28 | 0.55 |
| **Marital status** | | | | | | |
| Married | | | Reference | | | |
| Separated | 0.95 (.06) | 0.83–1.08 | 0.46 | 0.95 (.06) | 0.84–1.08 | 0.45 |
| Widow | 1.06 (.08) | 0.91–1.23 | 0.43 | 0.93 (.06) | 0.81–1.07 | 0.32 |
| Never married | 1.02 (.06) | 0.90–1.14 | 0.79 | 0.96 (.05) | 0.85–1.08 | 0.46 |
| **Study site** | | | | | | |
| Kiswa HC | | | **Reference** | | | |
| Komamboga HC | 0.68 (.07) | 0.55–0.85 | **0.001*** | 0.83 (.08) | 0.69–1.01 | 0.06 |
| Kitebi HC | 0.96 (.06) | 0.84–1.10 | 0.59 | 0.99 (.06) | 0.87–1.14 | 0.95 |
| Kasangati HC | 1.19 (.05) | 1.07–1.34 | **0.002*** | 1.16 (.07) | 1.02–1.32 | **0.02*** |
| Kisenyi HC | 0.88 (.05) | 0.77–0.99 | **0.04*** | 0.91 (.06) | 0.80–1.04 | 0.17 |
| Kawaala HC | 0.72 (.05) | 0.61–0.84 | **0.001*** | 1.01 (.09) | 0.84–1.20 | 0.93 |
| **Gender** | | | | | | |
| Male | | | | | | |
| Female | 1.16 (.06) | 1.05–1.29 | **0.005*** | 1.17 (.06) | 1.05–1.29 | **0.004*** |
| **Employment** | | | | | | |
| Formal | | | Reference | | | |
| Self | 0.82 (.03) | 0.75–0.91 | **0.001*** | 0.93 (.04) | 0.84–1.03 | 0.15 |
| unemployed | 0.68 (.06) | 0.58–0.82 | **0.001*** | 0.83 (.07) | 0.69–0.99 | **0.04*** |
| Informal | 0.64 (.06) | 0.53–0.77 | **0.001*** | 0.88 (.08) | 0.73–1.06 | 0.18 |
| **Seriousness of COVID-19** | | | | | | |
| Not serious | | | Reference | | | |
| Serious | 1.06 (.06) | 0.94–1.19 | 0.33 | 0.97 (.05) | 0.87–1.08 | 0.65 |
| **Perceived Risk** | | | | | | |
| Slight | | | Reference | | | |
| Very likely | 1.01 (.04) | 0.92–1.12 | 0.79 | 0.99 (.05) | 0.89–1.09 | 0.84 |
| **Personal health threat** | | | | | | |
| Minor | | | Reference | | | |
| Major | 1.19 (.05) | 1.09–1.31 | **0.001*** | 0.99 (.04) | 0.89–1.09 | 0.82 |
| **Prior COVID-19 testing** | | | | | | |
| No | | | Reference | | | |
| Yes | 1.27 (.05) | 1.16–1.39 | **0.001*** | 1.19 (.11) | 1.09–1.29 | **0.001*** |
| **General belief in benefit of vaccines** | | | | | | |
| Disagree | | | Reference | | | |
| Agree | 1.62 (.24) | 1.21–2.18 | **0.001*** | 1.12 (.11) | 0.92–1.36 | 0.26 |

*(Continued)*

**Table 6.** (Continued)

| | Crude prevalence ratio (PR) | | | Adjusted prevalence ratio (PR) | | |
|---|---|---|---|---|---|---|
| | PR (SE) | 95% CI | p-value | PR (SE) | 95% CI | p-value |
| **Most Ugandans want to be vaccinated** | | | | | | |
| Disagree | | | Reference | | | |
| Agree | 1.01 (.08) | 0.85–1.19 | 0.91 | 0.93 (.04) | 0.85–1.01 | 0.08 |
| **COVID-19 vaccination among PLWH** | | | | | | |
| Harmful | | | **Reference** | | | |
| Beneficial | 1.54 (.16) | 1.25–1.89 | **0.001**[*] | 1.18 (.07) | 1.04–1.33 | **0.008**[*] |
| **COVID-19 vaccines in Uganda are safe for PLWH** | | | | | | |
| Not safe | | | Reference | | | |
| Safe | 1.86 (.2) | 1.49–2.32 | **0.001**[*] | 1.45 (.11) | 1.24–1.70 | **0.001**[*] |
| **Received information during clinic visit** | | | | | | |
| Yes | | | Reference | | | |
| No | 0.68 (.08) | 0.53–0.86 | **0.002**[*] | 0.93 (.08) | 0.77–1.11 | 0.42 |
| **Access to COVID-19 jabs** | | | | | | |
| Difficult | | | | | | |
| Easy | 2.09 (.23) | 1.65–2.64 | **0.001**[*] | 1.64 (.18) | 1.31–2.05 | **0.001**[*] |

Deviance goodness-of-fit = 318.0574 Prob > chi2(737) = 1.0000 Pearson goodness-of-fit = 223.0274Prob > chi2(737) = 1.0000

relatively large sample compared to prior studies in SSA. There are some study limitations related to the design and representation. The study design was cross-sectional and the findings could differ over time. The respondents were recruited from six public health facilities in an urban setting while seeking care and as such may not be representative of all PLWH in Uganda especially in rural areas. Additionally, random numbers were used to select respondents in the care waiting area, posing a risk of recruitment bias.

Our results show high vaccine acceptance among this cohort of PLWH, and was positively associated with greater vaccine confidence, and perceived easiness (convince) to obtained the vaccine. Health campaigns need to tailor messaging on the benefits of the vaccines as well as PLHIV concerns related to vaccine safety in HIV and ART use. Additionally, individually empowering health care professionals to be more explicit in advising for vaccination.

## Supporting information

**S1 Dataset.**
(ZIP)

## Acknowledgments

We thank Aidah Nanvuma, and Joseph Ssenkumba for supporting the project management, Edson Mwebesa for the statistical review and Monica Agena, Immaculate Muloni, and Andrew Ashaba who worked as research assistants.

## Author Contributions

**Conceptualization:** Barbara Castelnuovo.

**Data curation:** Richard Muhindo, Agnes Kiragga.

**Formal analysis:** Agnes Kiragga.

**Funding acquisition:** Barbara Castelnuovo.

**Investigation:** Richard Muhindo.

**Methodology:** Richard Muhindo, Stephen Okoboi, Rachel King, Barbara Castelnuovo.

**Visualization:** Stephen Okoboi.

**Writing – original draft:** Richard Muhindo.

**Writing – review & editing:** Stephen Okoboi, Agnes Kiragga, Rachel King, Walter Joseph Arinaitwe, Barbara Castelnuovo.

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
