## [Decision Letter · Decision Letter 0]

2 Oct 2022

PONE-D-22-15743COVID-19 Vaccine Acceptability Among People Living with HIV in UgandaPLOS ONE

Dear Dr. Muhindo,

Thank you for submitting your manuscript to PLOS ONE. After careful consideration, we feel that it has merit but does not fully meet PLOS ONE’s publication criteria as it currently stands. Therefore, we invite you to submit a revised version of the manuscript that addresses the points raised during the review process.

 Your submission has been evaluated by three reviewers, their comments are included below. They raised a number of reporting issues, including requests for additional information about the questionnaire, sample size calculation, and results. Please address all of the reviewers' comments in revisions to your manuscript. To elaborate on a point raised by multiple reviewers, the manuscript needs to report additional information  about the development and validation of the questionnaire. The Methods section discussing this should include primary references that informed the questionnaire: currently you cite reference 24, a WHO website, but no specific research studies, scales or survey tools, or other resources that were used in developing questionnaire items. Additionally, please report in detail steps taken to translate the survey, to evaluate the validity and reliability of each version (language) of the questionnaire used in the study, and to evaluate/compare the consistency of the different translations used. Unless the questionnaire includes copyright-protected questions, a blank copy of the questionnaire should be provided as a Supporting Information file. Please also ensure that your Methods section provides a comprehensive description of inclusion and exclusion criteria that applied when selecting participants for the study. As a minor point, please use the same abbreviation for people living with HIV (either PLWHIV or PLWH) throughout the manuscript. 

We look forward to receiving your revised manuscript.

Kind regards,

Renee Hoch, Ph.D.

Managing Editor, PLOS Publication Ethics

PLOS ONE

Journal Requirements:

This study was supported through grant number D43TW009771 by National Institute for Health (NIH), Fogarty International Center.

Reviewers' comments:

Reviewer's Responses to Questions

**Comments to the Author**

1. Is the manuscript technically sound, and do the data support the conclusions?

Reviewer #1: Yes

Reviewer #2: Yes

Reviewer #3: Yes

2. Has the statistical analysis been performed appropriately and rigorously? 

Reviewer #1: Yes

Reviewer #2: Yes

Reviewer #3: Yes

3. Have the authors made all data underlying the findings in their manuscript fully available?

Reviewer #1: No

Reviewer #2: Yes

Reviewer #3: Yes

4. Is the manuscript presented in an intelligible fashion and written in standard English?

Reviewer #1: Yes

Reviewer #2: Yes

Reviewer #3: Yes

5. Review Comments to the Author

Reviewer #1: The manuscript is relevant, provides interesting findings and presented in a scientifically plausible manner. However you can consider providing the following information to make it stronger;

In this study COVID-19 vaccine uptake seems to be a major primary outcome that shouldn’t remain silent in the title statement. You could consider modifying the title to include uptake.

Provide more information on the reliability of the questionnaire after pretesting among the 15 respondents. The manuscript would benefit from provision of the Cronbach’s alpha.

You mention that the questionnaire was administered in the local language for some participants. Was it translated prior to administration or translation was left to the discretion of the research assistant. Provide more information on this in the methods as it can affect consistency of results obtained.

Further elaborate how the sample size was portioned among the participating study sites to avoid bias of distribution. If it wasn’t considered then it should be mentioned among the limitations.

Reviewer #2: Dear authors, thank you for submitting this interesting paper in the field of COVID -19. Please kindly refer to my suggestions and comments for your consideration.

Abstract

The conclusion addressed only the recommendation. Please update to include the conclusion.

Background

This section was well written. However, you may want to consider some of the grammatical errors I corrected within the body of the manuscript.

Line 86-87, Low intentions…. as only 57.2% of PLWH were willing to receive a COVID-19 vaccine. You may consider the above edited lines compared to what you have in the manuscript.

Lines 88, 92 consider replacing “vaccination hesitancy” with “vaccine hesitancy”

Methods

How did you arrive at the sample size of 767 participants? It’s important the readers understand how the sample size was computed.

The authors need to tell us more about the questionnaire? How many sections? What was the content of each section? Cronbach’s alpha of the questionnaire?

Separate the measurement of variables from the statistical analysis sub-section. The new sub-section on measurement of variables should come before statistical analysis

Results

What was the response rate?

Line 174 ‘A third (255)’ Please rewrite and maintain consistency with other sentences.

Add the 95% CI for the prevalence of vaccine acceptability.

Discussion

Well written and comprehensive. You need to address the generalizability of this study’s findings.

Tables

Be consistent with the number of decimal places (dps) across all tables. I advise you maintain 1dp. Also include the full meaning of HC, PLHW and MOH as footnotes in tables where they occurred.

Reviewer #3: A very good manuscript and right on time. The title is appropriate and adequate. The objectives, questions, contexts and claims makes sense to the topic. The figures and tables are clearly presented. The conclusion is supportive of the result and data.

However,

o There are some grammatical errors in lines 126,39,40,98 & 99.

o There is need to justify the sampling technique used for the study.

o It was not included in your recruitment and data collection procedure if the researcher had access to the sampling frame

which could have been a potential source of bias and as well a limitation to the study.

o The first sentence in the background needs to be referenced.

6. PLOS authors have the option to publish the peer review history of their article (what does this mean?). If published, this will include your full peer review and any attached files.

Reviewer #1: **Yes: **Andrew Marvin Kanyike

Reviewer #2: No

Reviewer #3: **Yes: **Olu-Abiodun Oluwatosin.O

---

## [Author Response · Author response to Decision Letter 0]

22 Oct 2022

Dr.Renee Hoch

Managing Editor, PLOS Publication Ethics

PLOS ONE

October 3rd, 2022

Dear Hoch, 

Regarding Manuscript Submission ID PONE-D-22-15743

We are very grateful for editorial and reviewer comments provided by the editors and reviewers of our manuscript titled: COVID-19 Vaccine Acceptability Among People Living with HIV in Uganda". Reviewer comments are encouraging, and the reviewers appear to share our view that study findings are of public health importance. The suggestions offered by the reviewers have been greatly helpful. Please find below our point-by-point responses in italics, with new text in bold font: 

Reviewer Comments

Editor 

1. Thank you for stating the following financial disclosure: This study was supported through grant number D43TW009771 by National Institute for Health (NIH), Fogarty International Center. Please state what role the funders took in the study. If the funders had no role, please state: "The funders had no role in study design, data collection and analysis, decision to publish, or preparation of the manuscript." If this statement is not correct you must amend it as needed. Please include this amended Role of Funder statement in your cover letter; we will change the online submission form on your behalf.

Response: We thank the editor for this observation, and suggestion. We apologize for not being categorical on this. We have implemented the suggestion, stated both in the cover letter, and in manuscript under the statement on funding (page 13, lines 298-300).

2. We note that you have indicated that data from this study are available upon request. PLOS only allows data to be available upon request if there are legal or ethical restrictions on sharing data publicly. For more information on unacceptable data access restrictions, please see http://journals.plos.org/plosone/s/data-availability#loc-unacceptable-data-access-restrictions. In your revised cover letter, please address the following prompts:

a) If there are ethical or legal restrictions on sharing a de-identified data set, please explain them in detail (e.g., data contain potentially sensitive information, data are owned by a third-party organization, etc.) and who has imposed them (e.g., an ethics committee). Please also provide contact information for a data access committee, ethics committee, or other institutional body to which data requests may be sent. b) If there are no restrictions, please upload the minimal anonymized data set necessary to replicate your study findings as either Supporting Information files or to a stable, public repository and provide us with the relevant URLs, DOIs, or accession numbers. For a list of acceptable repositories, please see http://journals.plos.org/plosone/s/data-availability#loc-recommended-repositories. We will update your Data Availability statement on your behalf to reflect the information you provide.

Response: We acknowledge this observation. In this study, there are no ethical or legal restriction on sharing the data set. The data set will be available as per journal requirement.

3. Your ethics statement should only appear in the Methods section of your manuscript. If your ethics statement is written in any section besides the Methods, please move it to the Methods section and delete it from any other section. Please ensure that your ethics statement is included in your manuscript, as the ethics statement entered into the online submission form will not be published alongside your manuscript

Response: We thank the editor for this suggestion. The suggestion has been implemented, and the ethics statement in the declaration section has been deleted.

Reviewer 1

1. The manuscript is relevant, provides interesting findings and presented in a scientifically plausible manner. 

Response: We thank the reviewer for this encouraging observation that our manuscript was well-written.

2. In this study COVID-19 vaccine uptake seems to be a major primary outcome that shouldn’t remain silent in the title statement. You could consider modifying the title to include uptake.

Response: We acknowledge this observation. We have implemented the suggestion on the title page.

The title now states as: COVID-19 Vaccine Acceptability, and Uptake Among People Living with HIV in Uganda

3. Provide more information on the reliability of the questionnaire after pretesting among the 15 respondents. The manuscript would benefit from provision of the Cronbach’s alpha.

Response: We thank the reviewer and agree that the suggested additions would improve the manuscript. The text below has been included in the method section, under recruitment and data collection procedures (page 7, lines, 151-154).

The questionnaire consisted of 26 question items (Cronbach’s α coefficient, 0.79), that were used to assess the major independent variables. Five items to assess complacency (α = 0.67), Nine to assess perceived vaccine confidence (α = 0.74), one to assess willingness to vaccinate, and four to assess convenience (α = 0.43) 

4. You mention that the questionnaire was administered in the local language for some participants. Was it translated prior to administration or translation was left to the discretion of the research assistant. Provide more information on this in the methods as it can affect consistency of results obtained.

Response: We are sorry for not being clear on this. In this study, we did not translate the English questionnaire into local language. However, the questionnaire was administered by experienced native speakers, who practiced during the training and piloting the administration of the questionnaire in local language. The text below has been included (page 7, lines 156-161)

All RAs were native speakers with experience of administering English interviews in the local language. 

5. Further elaborate how the sample size was portioned among the participating study sites to avoid bias of distribution. If it wasn’t considered then it should be mentioned among the limitations.

Response: we thank the reviewer for this observation, and we are agreement. We have provided an elaborate explanation of sample size estimation and allocation in methods under the section on recruitment and data collection procedures. 

The text below has been included (page 6, lines 136-146):

To estimate the study sample size, we aimed to achieve a precision of 5%. We assumed vaccine acceptance of 50%, as no prior studies in the region had described PLWH with regard to COVID-19 vaccination. A total of 768 respondents were estimated (two-sided test at 95% level of significance, 5% margin of error, and a design effect of 2) using Cochran formula [26]. Altogether 40,228 PLHIV ≥ 15 years were enrolled in care at the study sites. Of these 4543 (11.3%) were at Komamboga, 11950 (29.7%) at Kisenyi, (14.1%) at Kiswa, 6802 (16.9%) at Kitebi, 8825 (21.9%) at Kawaala, and 2420 (6%) at Kasangati. Using proportion to size allocation, we enrolled respondents at each health centre, while waiting to be seen by the health care providers. 

Reviewer 2

1. Dear authors, thank you for submitting this interesting paper in the field of COVID -19. Please kindly refer to my suggestions and comments for your consideration.

Response: We thank the reviewer for this uplifting comment about our work. 

2. Abstract: The conclusion addressed only the recommendation. Please update to include the conclusion.

Response: We apologize for this oversight. We are also grateful for the grammatical suggestions to abstract.

The text below has been included to cater for the conclusion (page 3, lines 62-64). 

vaccine acceptance was high among this cohort of PLWH, and was positively associated with greater vaccine confidence, and perceived easiness (convince) to obtained the vaccine.

3. Background: This section was well written. However, you may want to consider some of the grammatical errors I corrected within the body of the manuscript.

Response: We thank the reviewer for the encouraging and uplifting comment. We are sorry about the grammatical errors, and grateful to reviewer for the suggestions, which we have adopted. 

4. Line 86-87, Low intentions…. as only 57.2% of PLWH were willing to receive a COVID-19 vaccine. You may consider the above edited lines compared to what you have in the manuscript

Response: We are grateful to the reviewer for this improvement to our sentence presentation. We have adopted the suggestion on page 4, lines 100

5. Lines 88, 92 consider replacing “vaccination hesitancy” with “vaccine hesitancy”

Response: We are thankful to the reviewer for this observation and suggestion. We have implemented the recommendation on page 5, lines 102, and 106.

Methods

6. How did you arrive at the sample size of 767 participants? It’s important the readers understand how the sample size was computed.

Response: We apologize for not being clear regarding sample size estimation. 

The text below has been included (page 6, lines 136-146):

To estimate the study sample size, we aimed to achieve a precision of 5%. We assumed vaccine acceptance of 50%, as no prior studies in the region had described PLWH with regard to COVID-19 vaccination. A total of 768 respondents were estimated (two-sided test at 95% level of significance, 5% margin of error, and a design effect of 2) using Cochran formula [26].

7. The authors need to tell us more about the questionnaire? How many sections? What was the content of each section? Cronbach’s alpha of the questionnaire?

Response: We thank the reviewer for this observation. We have provided a statement to detail this.

The text below has been included (page7, lines 163-167):

The questionnaire consisted of 26 question items (Cronbach’s α coefficient, 0.79), that were used to assess the major independent variables. Five items to assess complacency (α = 0.67), Nine to assess perceived vaccine confidence (α = 0.74), one to assess willingness to vaccinate, and four to assess convenience (α = 0.43) 

8. Separate the measurement of variables from the statistical analysis sub-section. The new sub-section on measurement of variables should come before statistical analysis

Response: We thank the reviewer for this recommendation. We have implemented by including a section on study variables (page 7, line 166)

Results

9. What was the response rate?

Response: We acknowledge this concern by the reviewer. Our understanding is response rate relates to self-administered survey questionnaire, usually sent out. Our questionnaires were interviewer-administered. While we administered 768 questionnaires (total estimated sample size), analysis was performed on 767 (99.87%).

The text below has been included (page 9, line 206):

 Analysis was performed on 767 PLWH (99.87%)

10. Line 174 ‘A third (255)’ Please rewrite and maintain consistency with other sentences.

Response: We apologize for this inconsistency. This has been rewritten as one-third (33.2%) on page 9, line 209.

11. Add the 95% CI for the prevalence of vaccine acceptability.

Response: We acknowledge this suggestion. One page 9, line 213 we provide 95% CI for uptake (69.6%,95% confidence interval [CI]: 66.3%-72.8%), and acceptability (72.7%, 95% CI: 66.6%-78.0%) on line 215 

Discussion

12. Well written and comprehensive. You need to address the generalizability of this study’s findings.

Response: We thank the reviewer for the uplifting comment. Regarding generalizability of the study findings, we recognize on page 13, line 309 that our findings are not representative of all PLWH in Uganda especially in rural areas given that we enrolled participants from urban clinics.

Tables

13. Be consistent with the number of decimal places (dps) across all tables. I advise you maintain 1dp. Also include the full meaning of HC, PLHW and MOH as footnotes in tables where they occurred.

Response: we are grateful to the reviewer for the observations and suggestion. We have implemented the changes page on pages 20, 21, 22, 23 and 24.

Reviewer #3: 

14. A very good manuscript and right on time. The title is appropriate and adequate. The objectives, questions, contexts and claims make sense to the topic. The figures and tables are clearly presented. The conclusion is supportive of the result and data. 

Response: We are grateful to the reviewer for this uplifting and encouraging comment. 

15. There are some grammatical errors in lines 126,39,40,98 & 99.

Response: We apologize for these grammatical errors. We have corrected errors.

16. There is need to justify the sampling technique used for the study. It was not included in your recruitment and data collection procedure if the researcher had access to the sampling frame which could have been a potential source of bias and as well a limitation to the study.

Response: We acknowledge this observation from the reviewer. We have also explained under the recruitment that over 70 PLWH received care at each study site, but we used random numbers on each recruitment day. Additionally, acknowledged this in the limitations (pages 6, lines 136-147, and page 13 lines 310)

17. The first sentence in the background needs to be referenced.

Response: We thank the reviewer for this observation. The references below have been added (page 4, line 80).

1. Murray, C.J.J.T.L., COVID-19 will continue but the end of the pandemic is near. 2022. 399(10323): p. 417-419.

2. Del Rio, C. and P.N.J.J. Malani, COVID-19 in 2022—The Beginning of the End or the End of the Beginning? 2022. 327(24): p. 2389-2390.

We thank the reviewers for their helpful comments which have improved the manuscript. We trust that the revised manuscript will be suitable for publication in Plos one, but are happy to consider further revisions, if necessary. 

Sincerely

Richard Muhindo, RN, BSN, MPH

Senior Lecturer, Department of Nursing, College of Health Sciences

Makerere University

Corresponding author: r.muhindo@yahoo.com

---

## [Editor Report · Decision Letter 1]

22 Nov 2022

COVID-19 Vaccine Acceptability, and Uptake Among People Living with HIV in Uganda

PONE-D-22-15743R1

Dear Dr. Muhindo,

We’re pleased to inform you that your manuscript has been judged scientifically suitable for publication and will be formally accepted for publication once it meets all outstanding technical requirements.

The authors have reflected the updates of the three reviewers and have responded to the comments from the editor. The revisions are well elaborated within the body of the manuscript. I also wish to disclose that I participated as a reviewer for the initial evaluation of this manuscript.

Kind regards,

Chidinma Ihuoma  Amuzie, MBBS, MPHFEP, FWACP

Guest Editor

PLOS ONE
---

## [Editor Report · Acceptance letter]

24 Nov 2022

PONE-D-22-15743R1 

COVID-19 Vaccine Acceptability, and Uptake Among People Living with HIV in Uganda 

Dear Dr. Muhindo:

I'm pleased to inform you that your manuscript has been deemed suitable for publication in PLOS ONE. Congratulations! Your manuscript is now with our production department. 

Kind regards, 

on behalf of

Dr. Chidinma Ihuoma Ihuoma Amuzie 

Guest Editor

PLOS ONE